# Antitumoral Activity of a CDK9 PROTAC Compound in HER2-Positive Breast Cancer

**DOI:** 10.3390/ijms23105476

**Published:** 2022-05-13

**Authors:** María del Mar Noblejas-López, Lucía Gandullo-Sánchez, Eva M. Galán-Moya, Raquel López-Rosa, David Tébar-García, Cristina Nieto-Jiménez, Mónica Gómez-Juárez, Miguel Burgos, Atanasio Pandiella, Alberto Ocaña

**Affiliations:** 1Translational Research Unit, Albacete University Hospital, 02008 Albacete, Spain; mariadelmar.noblejas@uclm.es (M.d.M.N.-L.); evamaria.galan@uclm.es (E.M.G.-M.); raquel.lrosa@uclm.es (R.L.-R.); david.tebar@uclm.es (D.T.-G.); mgomezj@sescam.jccm.es (M.G.-J.); mburgosloz@unav.es (M.B.); 2Centro Regional de Investigaciones Biomédicas (CRIB), Castilla-La Mancha University (UCLM), 02008 Albacete, Spain; 3Instituto de Biología Molecular y Celular del Cáncer, CSIC, IBSAL and CIBERONC, 37007 Salamanca, Spain; lgandullo@usal.es (L.G.-S.); atanasio@usal.es (A.P.); 4Faculty of Nursing, Castilla-La Mancha University (UCLM), 02008 Albacete, Spain; 5Experimental Therapeutics Unit, Medical Oncology Department, Hospital Clínico San Carlos (HCSC), Instituto de Investigación Sanitaria (IdISSC) and CIBERONC, 28040 Madrid, Spain; cnietoj@salud.madrid.org; 6Department of Nutrition, Food Science and Physiology, School of Pharmacy and Nutrition, University of Navarra, 31008 Pamplona, Spain

**Keywords:** cyclin-dependent kinases, CDK9, HER2+ breast cancer, PROTACs, THAL-SNS-032

## Abstract

Cyclin-dependent kinases (CDKs) are a broad family of proteins involved in the cell cycle and transcriptional regulation. In this article, we explore the antitumoral activity of a novel proteolysis-targeting chimera (PROTAC) compound against CDK9. Breast cancer cell lines from different subtypes were used. Transcriptomic mapping of CDKs in breast cancer demonstrated that the expression of CDK9 predicted a detrimental outcome in basal-like tumors (HR = 1.51, CI = 1.08–2.11, *p* = 0.015) and, particularly, in the luminal B subtype with HER2+ expression (HR = 1.82, CI = 1.17–2.82, *p* = 0.0069). The novel CDK9 PROTAC, THAL-SNS-032, displayed a profound inhibitory activity in MCF7, T47D, and BT474 cells, with less effect in SKBR3, HCC1569, HCC1954, MDA-MB-231, HS578T, and BT549 cells. The three cell lines with HER2 overexpression and no presence of ER, SKBR3, HCC1569, and HCC1954 displayed an EC50 three times higher compared to ER-positive and dual ER/HER2-positive cell lines. BT474-derived trastuzumab-resistant cell lines displayed a particular sensitivity to THAL-SNS-032. Western blot analyses showed that THAL-SNS-032 caused a decrease in CDK9 levels in BT474, BT474-RH, and BT474-TDM1R cells, and a significant increase in apoptosis. Experiments in animals demonstrated an inverse therapeutic index of THAL-SNS-032, with doses in the nontherapeutic and toxic range. The identified toxicity was mainly due to an on-target off-tumor effect of the compound in the gastrointestinal epithelium. In summary, the potent and efficient antitumoral properties of the CDK9 PROTAC THAL-SNS-032 opens the possibility of using this type of compound in breast cancer only if specifically delivered to cancer cells, particularly in ER/HER2-positive and HER2-resistant tumors.

## 1. Introduction

Cyclin-dependent kinases (CDKs) are a broad family of proteins involved in the cell cycle and transcriptional regulation [1]. CDKs contain a kinase domain that, when bound to a cyclin, induces its activation by the phosphorylation of certain serines and threonines [2]. The deregulation of the activity of most CDKs has been linked to the oncogenic transformation and, therefore, has been considered as a potential druggable vulnerability [3]. Moreover, the presence of an enzymatic activity has made them attractive targets for the development of chemical small molecule inhibitors [4]. Of note, some of these compounds have reached the clinical setting, such as those acting on CDK4/6, which have shown to improve the outcome in hormone receptor-positive breast cancer [5,6]. Others are at different stages of drug development in different indications [7].

Transcriptional CDKs have been linked to cancer as they contribute to the transcription of genes that are considered as oncogenic transcription factors (TF), such as c-MYC [8,9]. CDK9 controls transcription by forming a protein complex called TAK/P-TEFb that phosphorylates the largest subunit of the RNA polymerase II [8,9]. Since the early use of the unspecific inhibitor flavopiridol, CDK9 inhibitors have been in preclinical and clinical development [10,11,12,13]. With the main limitation of a potential narrow therapeutic index, some compounds have reached late-stage clinical development. Increasing the potency and specificity has been the goal to optimize their development, reducing their side effects. Of note, optimization of the treatment schedule could be an alternative option to improve the safety profile, as has been the case for some novel chemical entities [14]. Another approach to target this protein is the development of Proteolysis Targeting Chimera compounds (PROTACs). One of these compounds, THAL-SNS-032, designed using the CDK9 inhibitor SNS-032 linked to a thalidomide derivative that binds to the E3 ubiquitin ligase cereblon, causes the degradation of CDK9 [15]. Although this compound has demonstrated activity in hematologic malignancies, little is known about the anti-tumoral activity of THAL-SNS-032 in different solid tumors including breast cancer, and about the toxicity profile and therapeutic index using in vivo models, as data in animals have not been reported [14].

Breast cancer is still an incurable disease in advanced stages. Indeed, even in some specific subtypes where recent treatment approvals have shown to improve outcome, the development of resistance to current therapeutics remains the main clinical problem [16,17]. In the case of HER2-positive breast cancer, resistance to trastuzumab or other anti-HER2 therapies often occurs over time, and strategies to overcome this resistance are needed [18,19,20]. In this context, the identification of mechanisms of resistance to anti-HER2 therapies in estrogen receptor (ER)-positive or -negative tumors is a main goal.

In this article, we describe the antitumoral activity of the CDK9 PROTAC THAL-SNS-032 in double HER2 and ER-positive breast cancer. Moreover, we report the antitumoral activity of that compound in cells with acquired resistance to several anti-HER2 therapies.

## 2. Results

### 2.1. In Silico Analysis of CDKs in Breast Cancer

To gain insights into the role of CDKs in breast cancer, we first explored the association between the expression of different CDKs and patient outcomes in the four described breast cancer molecular subtypes (basal-like, HER2+, Luminal A, and Luminal B with Status HER2+). To do so, we used the KM Plotter Online Tool that correlates the expression of 30k genes (mRNA level) and survival in more than 25 k samples from 21 tumor types including breast cancer. Sources for the databases include GEO, EGA, and TCGA. The CDKs evaluated included not only those involved in cell cycle regulation, CDK1, CDK2, CDK4, and CDK6, but also CDKs implicated in the control and regulation of transcription, CDK7, CDK8, CDK9, CDK12, and CDK13. In basal-like tumors, the expressions of CDK7, CDK8, and CDK9 were associated with worse survival; in HER2-positive tumors, those associated with a detrimental prognosis were CDK13, CDK7, CDK4, and CDK1. CDK8, CDK7, CDK4, and CDK1 predicted unfavorable outcomes in luminal A breast cancer patients. Lastly, CDK4, CDK7, and CDK9 predicted detrimental survival in the luminal B subtype. The *p*-value derived from the logrank test for each CDK in each breast cancer subtype is represented on the y-axis versus Hazard ratio on the x-axis (Figure 1A).

Among all these CDKs, we focused on CDK9, as kinase inhibitors against this protein have been used to develop novel proteolysis-targeting chimeras (PROTACs), and no assessment of antitumoral activity in breast cancer has been performed with this family of agents [14,21]. Of note, the expression of CDK9 predicted a detrimental outcome in basal-like tumors (HR = 1.51, CI = 1.08–2.11, *p* = 0.015) and, particularly in the luminal B subtype (HR = 1.82, CI = 1.17–2.82, *p* = 0.0069), a breast cancer tumor characterized by the double expression of ER and HER2. In line with the data shown in Figure 1A, in the HER2-positive subgroup, a trend toward worse outcome was observed, although it did not reach a statistically significant level (*p* = 0.053) (Figure 1B).

### 2.2. THAL-SNS-032 Is Active in HER2-Positive Cell Lines

The antitumoral activity of a novel CDK9 PROTAC termed THAL-SNS-032 was then explored in a panel of breast cancer cell lines representative of different molecular subgroups. We used MDA-MB-231, HS578T, BT549, SUM149, and HCC3153 as triple-negative (TN) breast cancer cell lines (estrogen receptor (ER)-negative, HER2-negative); SKBR3, HCC1569, and HCC1954 as HER2-positive cell lines (ER-negative, HER2-positive); MCF7 and T47D as a luminal A model (ER-positive, HER2-negative); and finally, BT474 as a luminal B example (ER- and HER2-positive). BRCA-mutated cell lines included SUM149, HCC3153, and HCC1954. Previously, we evaluated CDK9 protein levels in all cell lines by Western blot. CDK9 was increased in luminal A and luminal B in comparison with TNBC and HER2+ subtypes (Figure 2A,B). However, we verified CRBN and CDK9 expression levels in the described cells using the Cancer Cell Line Encyclopedia database. As shown in the Appendix A, respectively, no significant differences were observed between the molecular subtypes. The administration of THAL-SNS-032 displayed a deep inhibitory activity in luminal A and B cells MCF7, T47D, and BT474, as shown in Figure 2C, with EC50 doses below 100 nM. On the other hand, HER2+ and TN models, SKBR3, HCC1569, HCC1954, MDA-MB-231, HS578T, and BT549, needed higher concentrations to reach the EC50 (Figure 2D). In this context, the CDK9 protein level correlated with THAL-SNS-032 sensitivity. In contrast, the BRCA-status was independent to THAL-SNS-032 sensitivity.

Then, we also included BT474-derived cell lines representative of the adaptive resistance to trastuzumab (BT474-RH), to the antibody drug conjugate TDM1 (BT474-TDM1R), or to the kinase inhibitor lapatinib (BT474-LAPA-R) (Appendix A). As shown in Figure 2C, all these cell lines displayed a particular sensitivity to the CDK9 PROTAC THAL-SNS-032. An interesting finding was the fact that the three cell lines with HER2 overexpression and no presence of ER, SKBR3, HCC1569, and HCC1954 displayed an EC50 three times higher compared to ER-positive and dual ER/HER2-positive cell lines (Figure 2C). In luminal A, but specifically in BT474 and the resistant cells, doses between 50 and 75 nM reached the EC50, demonstrating that this group of cell lines display a higher sensitivity (Figure 2D).

### 2.3. THAL-SNS-032 Is More Potent Than the Parental CDK9 Kinase Inhibitor

The efficacy and potency of the CDK9 PROTAC was next compared to those of the parental kinase inhibitor SNS-032. As shown, THAL-SNS-032 was more potent than the kinase inhibitor SNS-032 (Figure 2E).

In addition to these standard cell culture studies, the efficacy of both drugs was also evaluated and compared in a Matrigel matrix-embedded 3D structures assay. These experiments demonstrated that THAL-SNS-032 and SNS-032 inhibited 3D structures formation in BT474, BT474-RH, and TDM1R cells (Figure 3A). However, the effect of the PROTAC was higher than that of SNS-032.

Following that, the effect of both drugs on cell adhesion was also evaluated. As can be seen in Figure 3B, the adhesion capacity of BT474, BT474-RH, and BT474-TDM1R was equally inhibited by the two compounds. Of note, the adhesion of the TDM1-resistant cells was less affected by these drugs.

Western blot analyses showed that THAL-SNS-032 caused a decline in CDK9 in BT474, BT474-RH, and BT474-TDM1R cells, but did not affect the transcriptional CDK7 levels, or the cell cycle regulatory CDK1 and CDK2 levels. In contrast, the parental SNS-032 CDK inhibitor did not affect the amount of any of the analyzed kinases in the three cell lines (Figure 3C).

### 2.4. Effect of THAL-SNS-032 on the Cell Cycle

To explore the mechanism by which THAL-SNS-032 affects viability, we first performed cell cycle analyses of the more sensitive cell lines. BT474, BT474-RH, and TDM1R were treated with SNS-032 or THAL-SNS-032 and, after 24 h, stained with propidium iodide. When evaluating the cell cycle phases using that experimental setting, no major changes were observed with either compound, the kinase inhibitor, and the CDK9 PROTAC, except in the case of a slight increase in G2 with THAL-SNS-032 in TDM1R (Figure 4A and Appendix A Appendix A). THAL-SNS-032 was able to reduce the levels of p21 more clearly in the TDM1R cell line. No changes were detected with SNS-032. A reduction in WEE1 was observed for both compounds in TMD1R cells, slightly observed in RH. Finally, an interesting finding was the fact that cyclin B was not present of TDM1R in comparison with the trastuzumab resistant and parental BT474 cell lines (Figure 4B). As no clear modifications in any phase of the cell cycle were identified after the administration of SNS-032 or THAL-SNS-032, we therefore inferred that the mechanism of action did not involve cell cycle arrest. To further elaborate on this, we proceeded to evaluate cell death induction.

### 2.5. THAL-SNS-032 Induces Cell Death

Next, we conducted assays for the evaluation of apoptosis. Thus, BT474, BT474-RH, and TDM1R were treated with SNS-032 or THAL-SNS-032 and, 72 h later, stained with Annexin V and Propidium iodide to detect apoptotic cells. These tests showed that SNS-032 had no appreciable effect on apoptosis. Conversely, THAL-SNS-032 caused a significant increase in the apoptotic populations in the three cell lines (Figure 5A and Appendix A Appendix A). In BT474-derived cell lines representative of adaptive resistance, the double-stained population (late apoptotic) increased compared with BT474. Western blot analyses indicated that, indeed, THAL-SNS-032 increased the levels of pH2AX, an indicator of DNA damage, while SNS-032 did not affect the levels of that protein. An insubstantial cleavage of PARP was observed in the three cell lines treated with either THAL-SNS-032 or SNS-032, even though a decrease in total PARP was observed in the cells upon treatment with these drugs. A slight reduction in caspase 3 was observed with the different treatments, which was more evident with the PROTAC in TDM1R (Figure 5B). In this line, THAL-SNS-032 activated caspase 3 in BT474, RH, and TDM1R cells, with mainlyin as the last one (Appendix A Appendix A). In this same cell line, the MCL1s isoform was reduced after treatment with THAL-SNS-032, with no significant modifications in the other cell lines. Administration of a broad-spectrum caspase inhibitor (QVD) reverted partially apoptosis in all models, suggesting that the mechanism was partially caspase-dependent (Appendix A Appendix A). Additional Western blot analyses showed no major changes in other caspases such as caspase 8 and caspase 9. We observed a slight reduction in pro-apoptotic family proteins BCL2 and BCLXL, and XIAP (inhibitor of apoptosis proteins family) after THAL-SNS-032 treatment. The same slight reduction was observed in pRNA polII and RNA polII (Appendix A Appendix A). These findings clearly demonstrate that the main effect of THAL-SNS-032 is mediated by the induction of apoptosis partially induced by caspases (Figure 5B).

### 2.6. THAL-SNS-032 Has an Inverse Therapeutic Index

We next explored the toxicity profile of SNS-032 and its PROTAC, THAL-SNS-032, in the normal epithelial breast cancer cell line MCF10A. As can be seen in Figure 6A, doses between 25 and 50 nM did not affect cell viability. However, doses of 100 nM significantly reduced proliferation for both agents and particularly with THAL-SNS-032, suggesting an inverse therapeutic index.

In this context, we explored different doses and schedules of administration for THAL-SNS-032, as can be seen in Figure 6B, in an exploratory safety study in immunosuppressed mice. Even using doses much below the potential toxic range, we observed that all the doses explored resulted to be toxic. In addition, no signs of tumor regression were observed (Figure 6C). The main toxicity observed included gastrointestinal disorders that were evident with physical and visual inspection, in addition to weight lost more prominently at the 10 mg/kg and 5 mg/kg level (Figure 6D and Appendix A Appendix A). The lowest dose included in the study (2.5 mg/kg every 3 days) was more than 10 times lower than the doses used with SNS-032 [22]. On the other hand, Western blot analysis of these tumors indicated a partial reduction of CDKs in the in vivo studies (Figure 6E and Appendix A Appendix A).

We used a bioinformatic approach to help predict the toxicity profile when these genes are absent, which could be extrapolated as the toxicity produced by the degradation of the protein with a PROTAC. Using the DepMap online tool, we observed how the elimination of CDK9, CDK7, and CDK1 with CRISPR impaired cell viability. Of note, the absence of both CDK9, CDK7, and CDK1 induced cell lethality and could be incompatible with cell survival, suggesting an inverse therapeutic index (Figure 6F).

In all genes, CRISPR knockout produced much stronger effects than partial reduction by RNAi. Indeed, for CDK9 siRNA genomic downregulation, no dependency was observed, which could suggest that partial kinase inhibition could be feasible but not the total degradation of the protein. To validate this approach in our cell line models, small interference genomic inhibition of CDK9 was performed. A reduction in CDK9 mRNA expression was confirmed (Appendix A Appendix A), which resulted in a partial reduction in CDK9 protein (Appendix A Appendix A). However, cell proliferation evaluated as MTT metabolization was not modified after siRNA knockdown of CDK9 (Appendix A Appendix A), suggesting that the complete reduction in the gene expression or protein degradation is necessary to induce a clear antiproliferative effect.

When evaluating the presence of these CDKs in normal tissue, we observed that CDK9 was mainly present in the gastrointestinal epithelium, which confirmed the toxicity identified in animals. As can be seen in Figure 6G, the gastrointestinal epithelium is a principal location of CDK9, which is in line with the gastrointestinal toxicity observed in mice. Globally, these data indicate that targeting CDK9 with a PROTAC, eliminating its expression, is a toxic approach and has an inverse therapeutic index. In this context, we defined “Inverse therapeutic index” as no therapeutic index, due to the target inhibited as an essential gene. In the case of CDK9, the toxicity is mainly produced by its effect on the normal gastrointestinal epithelium.

## 3. Discussion

In the present article, we describe the activity of the CDK9 PROTAC THAL-SNS-032 on several in vitro cellular breast cancer models. We found evidence supporting that those cells expressing both ER and HER2 are sensitive to the action of the CDK9 PROTAC. Moreover, this chimeric version of the inhibitor is more potent than the original compound and, notably, it also exerts antitumoral effects on cells with secondary resistance to classical drugs used in the HER2-positive breast cancer clinic. Identification of novel vulnerabilities is a main goal in cancer and, particularly in luminal B tumors, where the coexistence of two pathways, the ER and HER2 oncoprotein, drives the disease, and in those tumors where anti-HER2 therapeutic strategies have failed after a given period of treatment.

Clinical studies have shown promising results for the treatment of HER2-positive tumors with the incorporation of different strategies, beyond the mere administration of trastuzumab. This has been the case for the development of kinase inhibitors such as lapatinib, tucatinib, neratinib, or, more recently, the antibody drug conjugates TDM1 and trastuzumab deruxtecan [23,24,25,26]. Unfortunately, although these strategies augment survival, tumors become resistant and additional therapeutic options for our patients are required.

Therapeutic strategies against CDK9 have shown to be clinically active with some agents reaching the clinical setting [27]. In breast cancer, in vivo studies demonstrated the efficacy of CDK9 kinase inhibitors through an induction of apoptosis [28]. In this context, CDK9 inhibitors are currently in clinical development in several hematologic malignancies such as myeloid leukemia or myelodysplastic syndrome [9,28]. On the other hand, some CDK9 agents, such as dinaciclib, failed in phase III trials in chronic lymphocytic leukemia due to its toxicity profile, suggesting that the therapeutic index of these agents could be narrow [7]. In addition, the lack of specificity of the kinase inhibitor augmented its toxicity profile.

THAL-SNS-032 is a CDK9 PROTAC formed by the conjugation of SNS-032, a CDK9 kinase inhibitor, with a thalidomide through a PEG linker [15]. These compounds have demonstrated activity in some forms of leukemia but no evaluation in breast cancer has been performed yet. In the present work, we demonstrate the antitumoral activity of THAL-SNS-032 in cell lines expressing ER or both ER and HER2, including luminal A cells. However, very low activity was observed in basal-like cell lines. An interesting finding was the profound activity identified in several HER2-positive resistant models.

In relation to the mechanism of action, a clear induction of apoptosis was the main mechanism by which the agent was able to induce its effect, data that are in line with other articles targeting CDK9 with kinase inhibitors [28].

The fact that the compound mediates its effect in ER and HER2-positive cell lines suggests that a reduction in the transcription of specific genes linked with these two pathways could be involved in the mechanism of action. To this regard, some articles have described the role of different TFs, such as MYB, as key modulators of this effect in ER-positive breast cancer and explained how CDK9 inhibitors can modulate their expression [29,30]. An important aspect is the one associated with the difference between kinase inhibition and target degradation. When using a PROTAC, not only the kinase activity is disrupted but also other potential functions of the protein, and this could also be related to the presence of a different toxicity profile.

In our article, low doses of the compound induced severe toxicity in animals without observing any sign of antitumoral activity. The bioinformatic mapping on the expression of CDK9, in addition to the inhibitory effect on cell survival, suggested that the inverse therapeutic index was probably produced by the degradation of CDK9 in the gastrointestinal epithelium. In addition, the total effective degradation of CDK9 in vivo was not observed when analyzing protein expression by Western blot. Very little information has been reported regarding the in vivo tolerability and activity of CDK9 PROTACs. One study explored the pharmacodynamic profile of one of these compounds, demonstrating target engagement by performing single-dose experiments in animals; however, no long-term safety was reported in those studies [31]. The analysis of these data suggests that the degradation of CDK9 is a good therapeutic approach only if it is produced efficiently at the tumoral level.

In this context, there are different strategies to reduce the on-target off-tumor toxicity profile, therefore increasing the therapeutic index and increasing the effectivity of on-target degradation. Among them, two can be clinically explored: (i) identification of ligases that could be present in specific tumor types and not in nontransformed tissues, and (ii) improving the delivery through the formulation of the compound in nanoparticles. An example is the synthesis of vectorized nanoparticles with antibodies selected to target proteins expressed in the tumor cell membrane. In this context, our group is now creating a trastuzumab-coated vectorized nanoparticle loaded with THAL-SNS-032. This approach is feasible as we have demonstrated before with the BET-PROTAC inhibitor MZ1 vectorized in a nanoparticle containing trastuzumab [32]. Finally, other options to reduce toxicity while maintaining the efficacy include the optimization of the treatment schedule, as suggested elsewhere [14].

## 4. Materials and Methods

### 4.1. Clinical Outcome Evaluation

KM Plotter Online Tool [33] (http://www.kmplot.com, accessed on 4 May 2021) shows the potential relationship between CDKs expression and clinical outcome in breast cancer subtypes (basal-like, HER2+, luminal A, and luminal B with status HER2+). Patients were separated automatically according to the best cutoff scores of CDKs expression (lowest *p* value) into “high expression” versus “low expression”. The auto best cutoff was selected, so all possible cutoff values between the lower and upper quartiles were computed, and the best performing threshold was used as a cutoff. Breast cancer patients included in each subtype were: 417 for basal-like, 952 for luminal A, 198 for HER2+, and 263 for luminal B with status HER2+. A hazard ratio of 1 means a lack of association, a hazard ratio greater than 1 suggests an unfavorable outcome, and a hazard ratio below 1 suggests a favorable outcome.

### 4.2. Cell Culture and Drugs

Breast cancer cell lines MDA-MB-231, HS578T, SUM149, MCF7, T47D, SKBR3, and BT474 were cultured in DMEM (Sigma-Aldrich, Saint Louis, MO, USA) and BT549, HCC3153, HCC1569, and HCC1954 were cultured in RPMI (Thermo Fisher, Waltham, MA, USA). BT474 cells resistant to trastuzumab (BT474-RH), TDM1 (BT474-TDM1R), or lapatinib (BT474-LAPA-R) were generated by continuous exposure to the respective drugs, as described previously [34,35,36]. Normal-breast cell line MCF10A was cultured in DMEM/F12 complemented with 5% HS, 0.5 mg/mL of hydrocortisone, 10 µg/mL of insulin, 1% non-essential amino acids, and 20 ng/mL of EGF.

All culture media were complemented with inactivated FBS (10%) and antibiotics (100 U/mL of penicillin and 100 mg/mL of streptomycin). Cells were maintained under standard conditions (37 °C, 5% CO_2_).

All cell lines used were provided by Drs. J. Losada and A. Balmain, who purchased them from the ATCC, in 2015. Cells authenticity was confirmed by STR analysis at the Salamanca University Hospital (molecular biology unit).

The CDK-inhibitor SNS-032 was purchased from MedChemExpress (Monmouth Junction, NJ, USA) and the CDK-PROTAC THAL-SNS-032 was purchased from Tocris Bioscience (Bio-Techne R&D Systems, S.LU, Minneapolis, MN, USA). TDM1 (trastuzumab-emtansine) and trastuzumab were purchased from a local pharmacy. Lapatinib was purchased from Selleckchem (Houston, TX, USA).

### 4.3. Proliferation Assay (MTT) and EC50 Value

Proliferation assays were performed using MTT (Sigma-Aldrich, Saint Louis, MO, USA) colorimetric assays. Cells were seeded in 48-well plates (10,000 cells/well) and, after 24 h, were treated with SNS-032 or THAL-SNS-032. Culture medium was replaced at 72 h with MTT solution (phenol red-free DMEM with MTT 0.5 μg/μL) for 45 min at 37 °C. DMSO was then added to solubilize the samples. Absorbances at 555 nm values were recorded in a spectrophotometer multiwell plate reader. A reference wavelength of 690 nm was used. EC50 values were obtained using GraphPad Prism 7.0.

### 4.4. Matrigel-3D Tumor Sphere-Forming Assay

Cells (10,000 cells/well) were seeded in 48-wells plates with a Matrigel matrix and were treated 24 h later with 100 nM of SNS-032 or THAL-SNS-032. An inverted microscope was used for observing the formation of 3D structures after 72 h and the diameters of structures were quantified with ImageJ software.

### 4.5. Fibronectin-Adhesion Assay

BT474, BT474-RH, and BT474-TDM1R (100,000 cells/well in p6 multiwell) treated for 24 h with 100 nM of SNS-032 or THAL-SNS-032 were collected and seeded in fibronectin-coated plates (10 µg/mL in PBS, 1 h, 37 °C). After one hour, cells were washed with PBS and stained with crystal violet (0.05%, 15 min) and dissolved later in acetic acid (10%, 15 min). Absorbance (A_570_) was acquired using a spectrophotometer multiwell plate reader.

### 4.6. Protein Expression Analysis: Western Blot

BT474, BT474-RH, and BT474-TDM1R cells (100,000 cells/well in p6 multiwell) were treated with 50 nM of SNS-032 or THAL-SNS-032. Cells were then washed with cold PBS and lysed in RIPA buffer containing Protease and Phosphatase Inhibitor Cocktail (1%) (Sigma-Aldrich, Saint Louis, MO, USA). Later, insoluble material was deleted by centrifugation (10,000× *g*, 10 min). A BCA assay was used to obtain the protein concentration (Thermo Fisher, Waltham, MA, USA).

For Western blot, 40–80 μg of protein was isolated using 8–15% sodium dodecyl sulfate polyacrylamide gels that were devolved to polyvinylidene difluoride membranes (Merck millipore, Darmstadt, Germany). Membranes were blocked in 1×Tris-buffered saline (TBS, 100 mM Tris, pH 7.5, 150 mM NaCl) with 0.05% Tween and 1% BSA (1 h, RT). Then, membranes were incubated with the primary antibodies (1×TBS containing 0.05% Tween, overnight) included in Appendix A Appendix A. Horseradish peroxidase-coupled secondary antibodies (anti-rabbit 1:10.000, or anti-mouse 1:5.000) were used to detected protein-bound primary antibodies (1×TBS containing 0.05% Tween with 5% milk, 30 min, RT). Protein bands were exposed using the ECL Plus Western Blotting Detection System (GE Healthcare, Buckinghamshire, UK).

### 4.7. Flow Cytometry Experiments

The FACSCanto™ II flow cytometer was used for the experiments and FACS Diva software for analyses.

BT474, BT474-RH, and BT474-TDM1R cells (100,000 cells/well in p6 multiwell) were collected after 24 h of treatment and fixed in ethanol (70%, ice-cold, 30 min). Cell pellets were washed in PBS with 2% BSA and incubated in the dark (1 h, 4 °C) with Propidium iodide/RNAse staining solution (Immunostep, Salamanca, Spain) to analyze the cell cycle.

BT474, BT474-RH, and BT474-TDM1R cells (100,000 cells/well in p6 multiwell) were collected after 72 h of treatment and stained for 1 h with Annexin V Binding Buffer containing Annexin V-DT-634 (AV) (Immunostep) and Propidium iodide (PI) (2 mg/mL) to analyze the cell death. Broad-spectrum inhibitor Q-VD-Oph (QVD) (10 μM, Sigma Aldrich) was added 45 min previous drug exposure.

### 4.8. Xenograft Mice (In Vivo)

We used 4–5-weeks-old female BALB*/c nu/nu* mice (*n* = 8). Mice were orthotopically injected with BT474 cells (5 × 10^6^) in PBS solution with 50% of Matrigel (Sigma-Aldrich, Burlington, MA, USA). Treatment was initiated when tumors arrived at a volume of about 100 mm^3^. Animals were separated into three doses and schedules of administration for THAL-SNS-032 and vehicle ((2-hydroxypropyl) Beta-cyclodextrina, 10%)-treated control (*n* = 2). THAL-SNS-032 was intraperitoneally injected at doses of: 10 mg/kg once a week, 5 mg/kg twice a week, and 2.5 mg/kg three times per week. Tumors were measured every three days. Tumor volumes were calculated using the formula: V = (L × W^2^)/2, where V = volume (mm^3^), L = length (mm), and W = width (mm). Mice were sacrificed by CO_2_ inhalation and tumors was stored at −80 °C.

For Western blotting, tumors were washed (PBS) and homogenized mechanically in ice-cold lysis buffer (1.5 mL/100 mg of tumor). This homogenate was centrifuged at 10,000× *g* for 20 min. Western blotting was performed as described previously.

### 4.9. Dependency Study and Cell Line Expression Analyses

The DepMap online tool [37,38] (https://depmap.org/portal/, accessed on 27 May 2021) was used to analyze the dependency of CDKs (cell cycle and transcriptional) of a panel of tumor cell lines. DepMap includes a library of human genes that has been knocked down or knocked out through CRISPR technology in large panels of human cell lines representing different types of cancer, followed by enrichment/depletion analysis of gene-specific shRNAs or sgRNAs. The probability of dependence of each cell line on the searched gene is plotted as dependence scores, where strongly negative values indicate those cell lines in which a specific gene is especially important for survival. The CRISPR (DepMap 21Q3 Public + Score, CERES) and Combines RNAi (Achilles + DRIVE + Marcotte, DEMETER2) databases were used (accessed on 27 May 2021).

The Cancer Cell Line Encyclopedia (https://sites.broadinstitute.org/ccle/, accessed on 29 November 2021) is a database of gene expression, genotype, and drug sensitivity data for human cancer cell lines [39]. We analyzed CRBN and CDK9 expression in breast cancer cell lines using this platform (Expression 21Q4 Public (accessed on 29 November 2021).

### 4.10. Small Interfering RNA CDK9

siRNA oligonucleotides CDK9 (EHU053481, Sigma-Aldrich, Saint Louis, MO, USA) were transfected into cells using the Lipofectamine RNAiMax protocol (Life Technologies, Carlsbad, CA, USA) at a final concentration of 20 nM. Briefly, cells (100,000 cells/well in p6 multiwell) were transfected (~80% of confluency); after 24 h, cells were reseeded for validation experiments (qPCR and Western blot) and the genomic down-regulation effect in proliferation was evaluated (Proliferation assay (MTT)).

### 4.11. Quantitative Reverse-Transcription PCR

RNeasy Mini Kit (Qiagen, Hilden, Germany) was used for RNA extraction. Determinations of the concentration and purity were performed using a NanoDrop ND-1000 spectrophotometer (Thermo Fisher Scientific, Waltham, MA, USA). An amount of 1 μg of total RNA was reverse-transcribed using the RevertAid H Minus first-strand cDNA synthesis kit (Thermo Fisher Scientific, Waltham, MA, USA.) in a thermal cycler (Bio-Rad, Hercules, CA, USA) under the reaction conditions: 65 °C for 5 min, 42 °C for 60 min, and 70 °C for 10 min. The StepOnePlus Real-Time PCR system (Applied Biosystems, Bedford, MA, USA) was used for real-time PCR analysis according to the manufacturer’s instructions. The primer sequences used were: h-CDK9 F: ACTTCTGCGAGCATGACCTT, h-CDK9 R: AAAGTCTGCCAGCTTCAGGA, h-18S F: GAGGATGAGGTGGAACGTGT, h-18S R: TCTTCAGTCGCTCCAGGTCT. At first, 95 °C for 5 min was performed. Later, 40 cycles of 95 °C for 15 s was performed and finished by 60 °C for 1 min. Each sample was analyzed in triplicate and cycle threshold (Ct) values of transcripts were determined using StepOne Software v.2.1. The 18S gene was used as the housekeeping gene. Untreated control cells were used as the control to determine the X-fold mRNA expression.

### 4.12. CDKs Expression in Normal Tissues

Expression levels of CDKs in normal tissues were analyzed using the GEPIA2 web server [40] (Gene Expression Profiling Interactive Analysis; http://gepia2.cancer-pku.cn/, accessed on 27 May 2021). GEPIA2 is an updated version of GEPIA for analyzing the RNA sequencing expression data of 9736 tumors and 8587 normal samples from the TCGA and the GTEx projects. The CDKs' median expression in normal samples is shown in a bodymap in Log2(Transcript Per Million + 1) Scale.

### 4.13. Caspase 3 Activity

Caspase reaction buffer was added to the protein extract (50 μg, 1 h, 37 °C, in the dark). Then, the fluorescence at 400 nm was measured in a spectrophotometer multiwell plate reader. A reference wavelength of 505 nm was used.

### 4.14. Statistical Analysis

All results are shown as the mean ± SEM of three independent experiments, each of them performed at least in triplicate. All data were analyzed using the statistical software GraphPad Prism 7.0. The *t*-test for independent samples or one-way ANOVA assay with the Tukey subtype was used. The level of significance was determined at 95% (*p* ≤ 0.05 = *; *p* ≤ 0.01 = **; *p* ≤ 0.001 = ***).

## 5. Conclusions

We describe a novel vulnerability of ER/HER2-positive tumors and HER2-resistant tumors. CDK9 degradation has a more potent antitumoral effect than kinase inhibition alone but the inverse therapeutic index limits its potential development. Options to specifically degrade CDK9 in the tumor should be pursued.

## Figures and Tables

**Figure 1 ijms-23-05476-f001:**
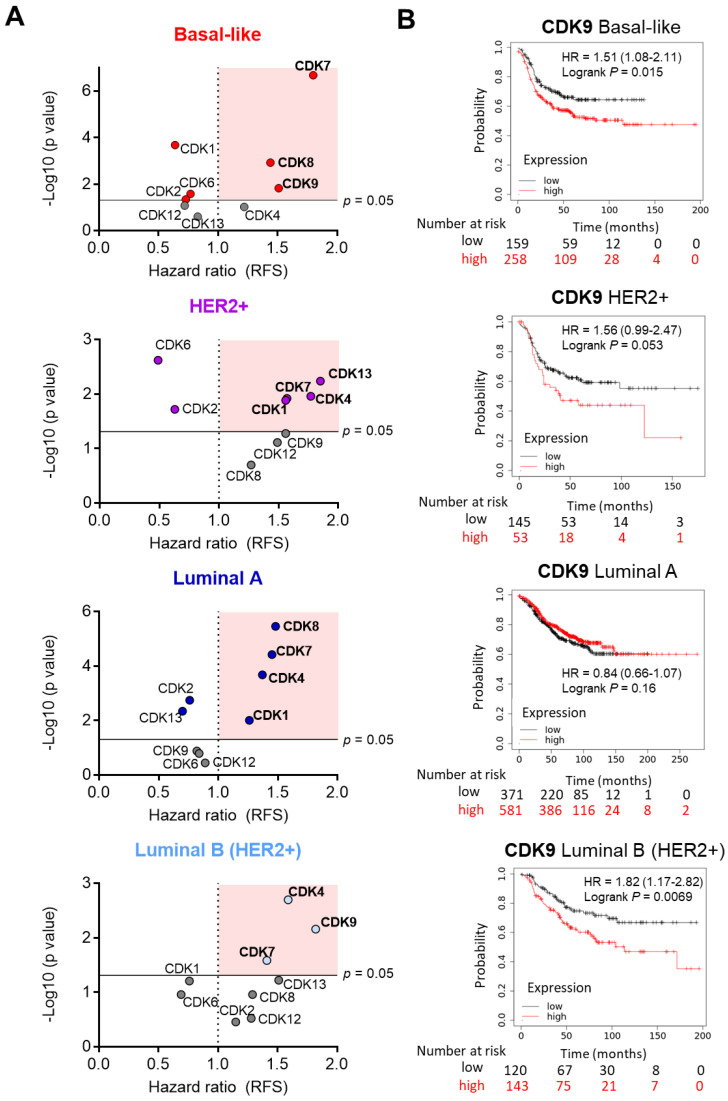
Correlation between CDKs expression level and clinical outcome in breast cancer. Graphs displaying HR values and *p*-value derived from the logrank test extracted from Kaplan–Meier survival plots of the association between CDKs individually expressed and patient prognosis, including relapse-free survival (RFS), for breast subtypes: basal-like (*n* = 417), Luminal A (*n* = 952), HER2+ (*n* = 198), and Luminal B with Status HER2+ (*n* = 263). CDKs with a significantly detrimental outcome association are included in highlighted square (**A**). Kaplan–Meier survival plots of the association between CDK9 mean expression levels and patient prognosis, including relapse-free survival (RFS) (*n* = 1764) for all breast subtypes: basal-like (*n* = 417), Luminal A (*n* = 952), HER2+ (*n* = 198), and Luminal B with Status HER2+ (*n* = 263) (**B**).

**Figure 2 ijms-23-05476-f002:**
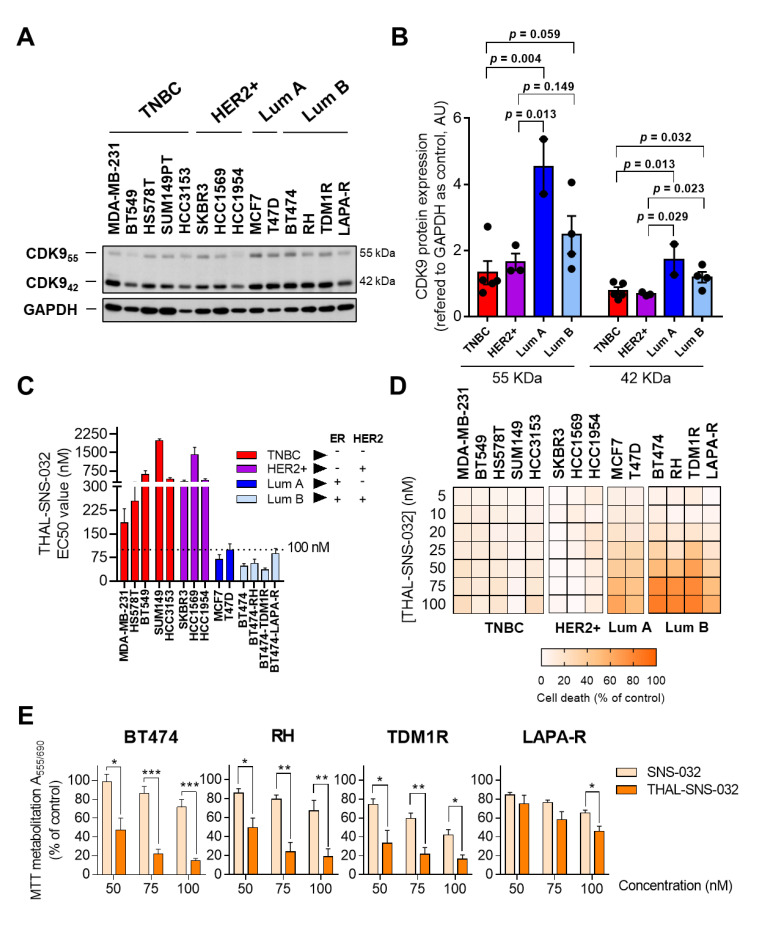
Evaluation of CDK9 protein expression and antiproliferative activity of SNS-032 and THAL-SNS-032 in breast cancer cell lines. Protein levels of CDK9 in all cell lines were analyzed by Western blot (**A**). Band quantification of CDK9 protein level referred to GAPDH as the loading control (**B**). THAL-SNS-032 EC50 obtained by MTT proliferation assays in breast cancer cell lines after 72 h of treatment (**C**). Heat map depicting the effect of different doses of THAL-SNS-032 in cell death in a breast cancer cell line panel (**D**). THAL-SNS-032 reduced cell viability more than SNS-032 in BT474, BT474-RH, BT474-TDM1R, and BT474-LAPA-R at the doses indicated (72 h). We used MTT assay (**E**). * *p* < 0.05; ** *p* < 0.01; *** *p* < 0.001.

**Figure 3 ijms-23-05476-f003:**
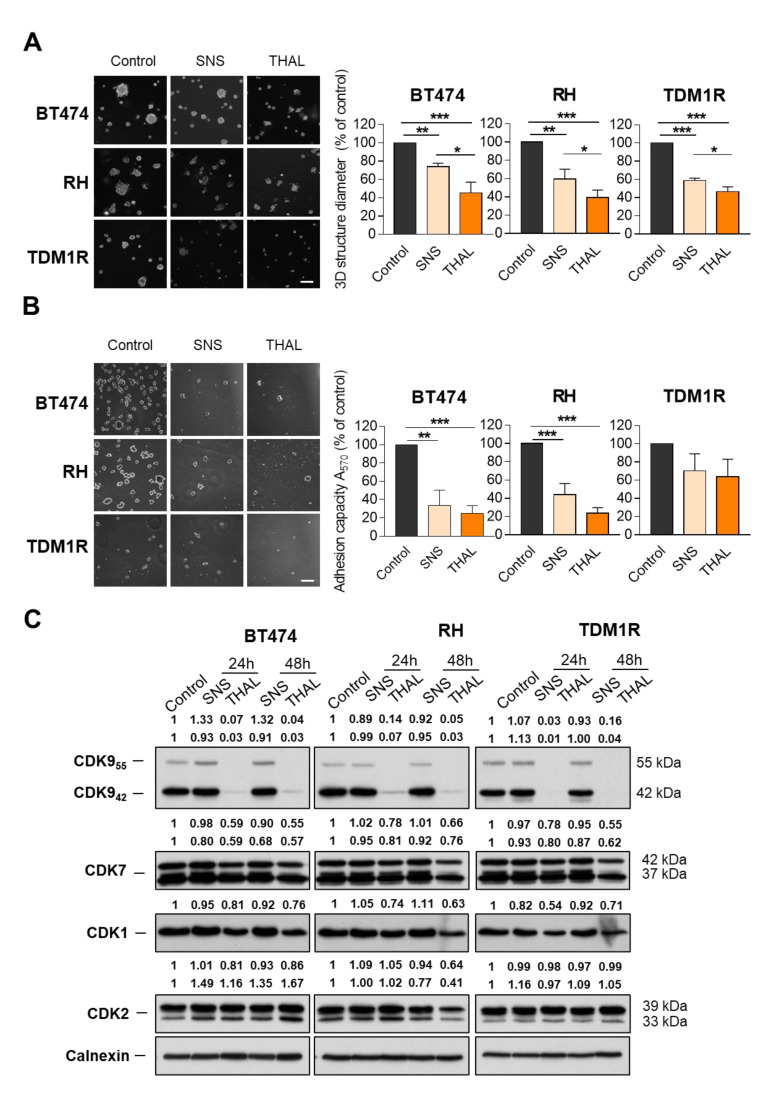
Antitumoral effect of SNS-032 and THAL-SNS-032 in 3D-matrix and adhesion assays in breast cancer lines and CDKs protein level evaluation after treatments. SNS-032 or THAL-SNS-032 (100 nM) was applied for 72 h to BT474, RH, and TDM1R cells seeded in a Matrigel matrix. Results presented refer to control cells. Scale bar = 100 μm (**A**). Cell adhesion to fibronectin substrate after 24 h of exposure to SNS-032 or THAL-SNS-032 (100 nM) (**B**). Expression levels of CDK9, CDK7, CDK1, and CDK2 in BT474, RH, and TDM1R cells treated with SNS-032 and THAL-SNS-032 at the times indicated at 50 nM. Scale bar = 100 μm (**C**). * *p* < 0.05; ** *p* < 0.01; *** *p* < 0.001.

**Figure 4 ijms-23-05476-f004:**
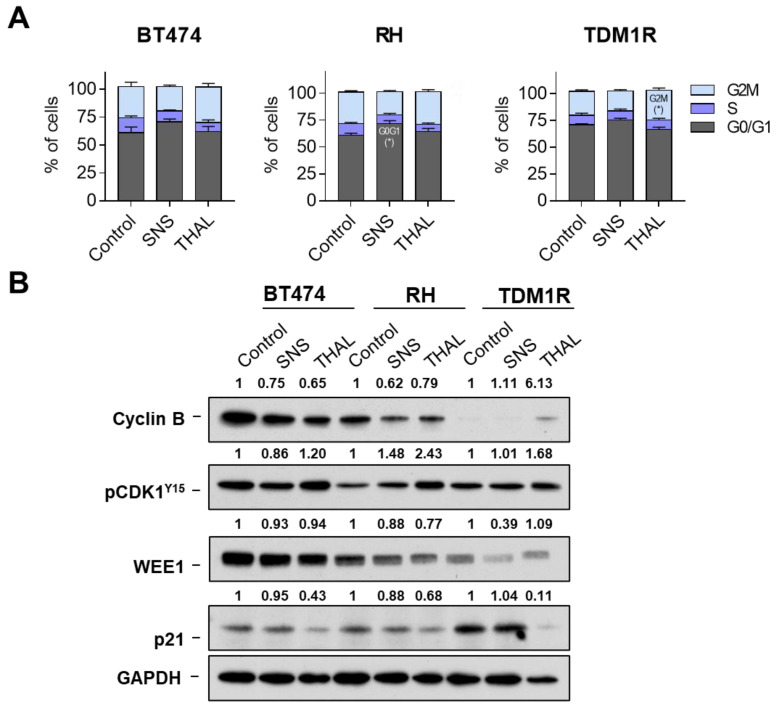
Cell cycle analyses of SNS-032 and THAL-SNS-032 in BT474 and BT474-derived cell lines representative of adaptive resistance. Bar graph showing populations generated by flow cytometry in each phase of cell cycle for BT474, RH, and TDM1R cells of SNS-032 (50 nM) and THAL-SNS-032 (50 nM) for 24 h (**A**). Expression of proteins involved in cell cycle in cell lines treated with SNS-032 (50 nM) and THAL-SNS-032 (50 nM) for 24 h. GAPDH was used as a loading control (**B**). * *p* < 0.05.

**Figure 5 ijms-23-05476-f005:**
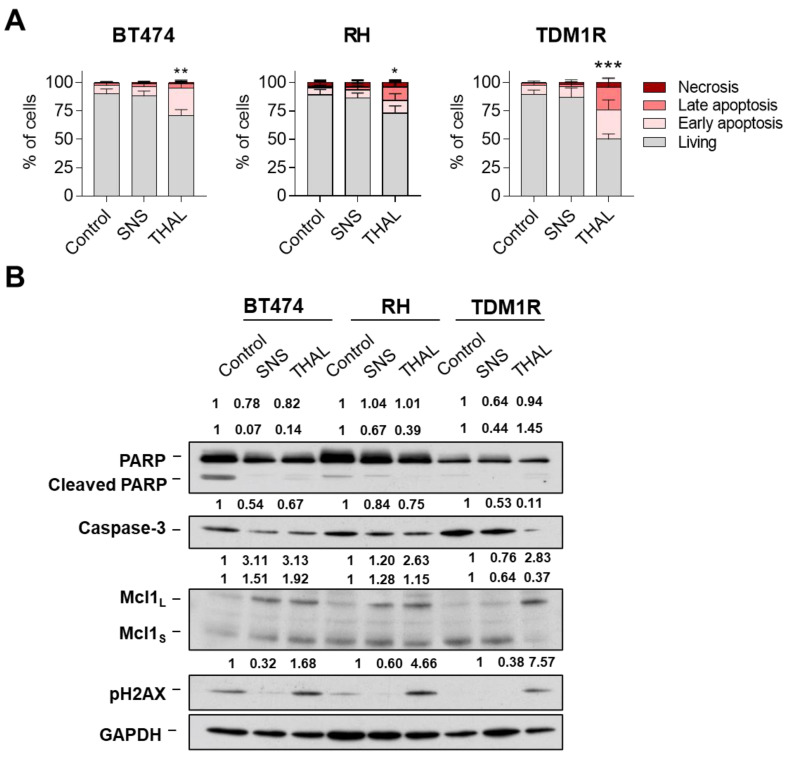
Apoptosis process of SNS-032 and THAL-SNS-032 in BT474 and BT474-derived cell lines representative of adaptive resistance. Cell death following SNS-032 (50 nM) or THAL-SNS-032 (50 nM) treatment after 72 h in BT474 and BT474-derived cell lines representative of adaptive resistance cell lines and resistant cell lines was evaluated by flow cytometry with annexin V and propidium iodide staining (**A**). Expression of proteins involved in cell death was evaluated by Western blot in cells lines treated at 50 nM (SNS-032 and THAL-SNS-032) for 72 h. GAPDH was used as a loading control (**B**). * *p* < 0.05; ** *p* < 0.01; *** *p* < 0.001.

**Figure 6 ijms-23-05476-f006:**
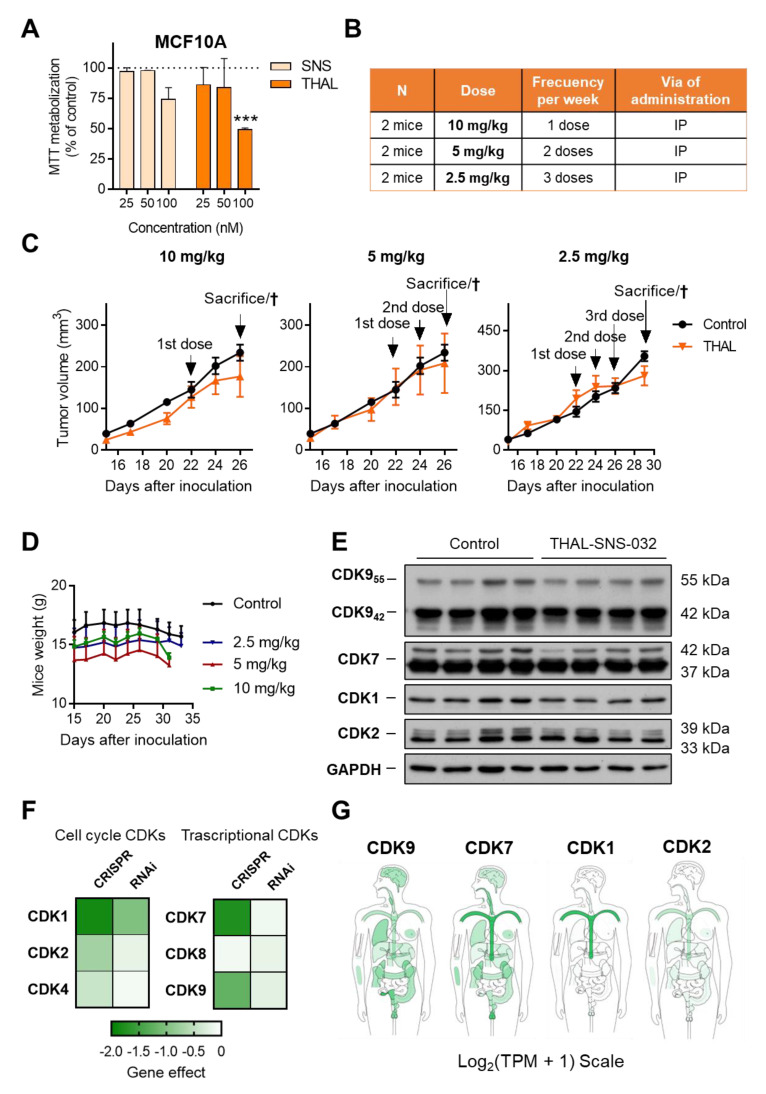
THAL-SNS-032 therapeutic index exploration. Normal breast cancer cell line MCF10A was treated with 25, 50, and 100 nM of SNS-032 and THAL-SNS-032. Proliferation was evaluated by MTT assay after 72 h of treatment (**A**). Table summary of THAL-SNS-032 doses, frequency, and route of administration in in vivo studies (**B**). Mice engrafted with BT474 cells were treated with THAL-SNS-032 (at indicated doses, I.P.) and controls with excipient. Mean of tumor volume ± SEM at each point was represented. Day of dose administration and sacrifice are indicated in each graph (**C**). Mice weight during the in vivo experiments. Treated mice (specially 10 mg/kg and 5 mg/kg) lost weight after THAL-SNS-032 administration (**D**). Western blot showing the expression levels of CDKs in tumors treated or untreated with THAL-SNS-032 (**E**). Heat map of the dependency score obtained by DepMap to evaluate CDKs dependency after CRISPR (DepMap 21Q3 Public + Score, CERES) or RNAi (Achilles + DRIVE + Marcotte, DEMETER2) inhibition (**F**). Body map showing expression of CDK9, CDK7, CDK1, and CDK2 in normal tissues in Log2(Transcript Per Million + 1) Scale. Adapted from: GEPIA2 web server (Gene Expression Profiling Interactive Analysis. http://gepia2.cancer-pku.cn/, accessed on 27 May 2021) (**G**). *** *p* < 0.001.

## Data Availability

Data used in this study are available in public functional genomics data repositories or are available from the corresponding author upon reasonable request.

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
