# Peer review of "Antitumoral Activity of a CDK9 PROTAC Compound in HER2-Positive Breast Cancer"

_ijms, 2022, doi:10.3390/ijms23105476_

Round 1
Reviewer 1 Report
Noblejas-Lopez et al. write about their efforts to target Cdk9 using a PROTAC strategy, in breast cancer models. This is the same work that was previously submitted to IJMS (1592832). I felt that the previous manuscript lacked value. The authors’ rebuttal seemed fair when they felt that the use of a PROTAC strategy on Cdk9 was a novel one despite Cdk9 being an established target of cancers. The issue is that the data is underwhelming. I respect the authors’ assertion in wanting to report the data as it is, but for this to be worth reading and citing when published would require the focus of the manuscript to be altered.
I am surprised to not find major changes in their resubmission version, and I strongly suggest a major rewrite. Below are my specific feedback points:
The pitch:
The introduction section should be written to clearly convey the focus as being on documenting the efficacy and toxicity of THAL-SNS-032. The writing should emphasize the value in efforts towards identifying a Cdk9-targeting regimen for cancers that would be effective and safe. This also calls for more elaboration in the discussion section of the manuscript as well.
The writing must include a discussion of comparison of the study’s strategy to other published PROTAC-based therapeutics that may be responsible for the toxicity profile. In addition, it must specifically also discuss literature on PROTAC-based Cdk9 targeting as well, for instance: https://pubmed.ncbi.nlm.nih.gov/33338869/
Results:
Data pertaining to Figures 4 and 5 are not very convincing in terms of the effects of THAL-SNS-032 on cell cycle and apoptosis. Figure 4B is just stated as a plain observation with no interpretations into it.
Grammar and word choice:
The writing must be tweaked at quite a few places to use the right words. Out of several, one instance is in lines 131 – 133: “sensibility” should be “sensitivity”.
Reviewer 2 Report
The authors addressed the previous critique satisfactorily.
Author Response
Response: We appreciate this comment.
Reviewer 3 Report
In the revised version, the authors of manuscript entitle “Antitumoral Activity of a CDK9 PROTAC Compound in HER2 Positive Breast Cancer” have addressed my main concerns.
Author Response
Response: We appreciate this comment.
Reviewer 4 Report
The authors addressed most of the points raised by the reviewers.
Author Response
Response: We appreciate this comment.
Round 2
Reviewer 1 Report
No further comments.
This manuscript is a resubmission of an earlier submission. The following is a list of the peer review reports and author responses from that submission.
Round 1
Reviewer 1 Report
The biggest issue with this study is the lack of novelty and significance of targeting Cdk9 in cancers. Cdk9 has extensively been demonstrated to be a worthy target of cancers (see among others: https://www.mdpi.com/2072-6694/13/9/2181 and https://www.frontiersin.org/articles/10.3389/fphar.2020.01230/full). However, as a double-edged sword, Cdk9 is also a very non-specific target, essential for normal tissue functioning. The real challenge is in designing effective therapeutics to this promising target, that would specifically act against cancer cells.
While the in vivo experiment and the simulation turned out to be negative, and while the candor of the authors is to be appreciated for reporting it as it is, it is not surprising either.
The authors would need to really think about the value of this piece of work and I urge them to re-package this with other unpublished data they may have to salvage this study into something valuable.
Reviewer 2 Report
This paper has a nice exploration of the activity of THAL-SNS-032 in various breast cancer cell lines that have been stratified into 4 subtypes. The conclusions of the cell data are clear and appropriate controls are used like the inclusion of the effect on CDK1, 2, and 7 as was done in the original Gray paper where this compound was first described. An interesting finding is the increased activity in some of the drug resistant cell lines. Further experiments to assess the MOA are appropriate and add value to the results. Even though the results of the animal study suggested that this drug is not a viable approach in vivo in this current format, the authors list some future work that may be able to mitigate this. The biggest issue was that some of the figures were hard to see/read. The subsequent comment are more minor.
Figures throughout – some of the labels are very hard to read at 100% size and are blurry if further zoomed in. Please make sure they meet the minimum font size, use a clear font and come through crisply.
Line 68 “therapiesoften occurs over time ,”; insert space between words and delete space before “,”.
Line 77-78 “To gain insights into the role of CDKs in breast cancer, we first explored the association between the expression of different CDKs and patient outcome” It would be helpful here to understand where the data used in the analysis was gathered from. In the materials and methods is talks about breast cancer patients, but not where the data comes from. One can go to KMPlot and try to ascertain this, but a quick sentence on how patient data is selected for inclusion may help understand how to interpret the results.
Line 100 “Among all these CDKs, we focused on CDK9, as kinase inhibitors against this protein have been used to develop novel proteolysis targeting chimeras (PROTACs)”, please add a reference here to explain what you are referring to. You could put your reference 14 here again for example to make it clear. For other PROTACs that you are referring to, please add additional references.
Line 170 “compoundsin” insert space between words.
Line 200 “Inverse Therapeutic Index” is not a particularly common phrase, I would consider replacing this. It is not 100% clear what is meant here by inverse vs. reverse vs. narrow therapeutic index (all 3 of these are used in different places to describe the results) and what the implications are for this being a viable breast cancer treatment strategy.
Line 297 “Without mentioning the complete presence of the protein and the long-lasting effect.” This is not a complete sentence.
Reviewer 3 Report
In this MS the authors have investigated the antitumoral activity of a CDK9 PROTAC THAL-SNS-032 against breast cell lines. They showed that the THAL-SNS-032 exhibited cytotoxic activity in HER2 positive cell lines and in drug-resistance cells. They indicated that apoptosis is the mechanism involved in this effect. They also evaluate the safety profile of this compound, showing the THAL-SNS-032 is cytotoxic for gastrointestinal epithelium. The paper is well written, however, following points are necessary to be considered by the authors.
- Authors should improve the resolution of figures.
- Please, include the densitometric analysis of western blot results, mainly for molecules with expression changes.
- Since authors suggest that apoptosis is the mechanism responsible for cytotoxic effect of PROTAC, could explain clearly why drug-resistance cells are more susceptible.
- Could authors discuss about the relation between cytotoxic effects on gastrointestinal epithelium observed and route of administration?
Reviewer 4 Report
The authors explore the anti-tumoral activity of a proteolysis targeting chimera (PROTAC) compound (THAL-SNS-032) against CDK9 in a panel of breast cancer cell lines. In vivo experiments show toxicity and lack of efficacy of THAL-SNS-032.
The authors tested THAL-SNS-032 on three triple-negative breast cancer (TNBC) cell lines, which do not include BRCA1-mutant cells. Given the effects of THAL-SNS-032 on DNA damage, it would be interesting to analyze the effects of this compound on BRCA1-mutant TNBC cells.
The authors analyzed the expression of CDK9 mRNA in the breast cancer cells which have been treated with THAL-SNS-032 IC50. However, the difference in IC50 observed amongst different cell lines might be ascribed to difference in CDK9 protein levels. Thus, CDK9 protein expression analysis should be performed.
In vitro experiments show that THAL-SNS-032 is more potent than SNS-032 alone, indicating kinase independent tumorigenic functions of CDK9. Have these been previously reported? This should be better discussed
Cell death analysis data shown here are not consistent with expression of apoptosis markers in response to THAL-SNS-032. How can these data be reconciled?
The authors state that in vivo treatment with THAL-SNS-032 leads to down-regulation of CDK9 in tumour cells. However, quantification of CDK9 protein expression does not show any significant difference in CDK9 protein levels. Conclusions should be revised accordingly in the text.
Did any previous studies report in vivo toxicity of THAL alone (without-SNS-032)? This is an important experimental arm missing in this study.